# Robotic Platforms for Therapeutic Flexible Endoscopy: A Literature Review

**DOI:** 10.3390/diagnostics14060595

**Published:** 2024-03-11

**Authors:** Naoya Tada, Kazuki Sumiyama

**Affiliations:** Department of Endoscopy, The Jikei University School of Medicine, Tokyo 105-8461, Japan; kaz_sum@jikei.ac.jp

**Keywords:** flexible robotic endoscopy, natural orifice translumenal endoscopic surgery, endoscopic submucosal dissection, endoscopic full-thickness resection

## Abstract

Flexible endoscopy, initially developed for diagnosis and tissue sampling, has been adapted for therapeutic interventions, leading to the emergence of natural orifice transluminal endoscopic surgery (NOTES) in the 2000s. The need for a triangulation function to enhance the intuitiveness and safety of NOTES has prompted the development of dual-arm, flexible endoscopic robotic platforms. Although the global interest in NOTES has decreased in the last decade, no-scar surgery concepts are still being applied to other complex endoluminal interventions, such as endoscopic submucosal dissection (ESD) and endoscopic full-thickness resection (EFTR), with ongoing research and development. The application of robotics in flexible endoscopy may facilitate the standardization of these procedures and expedite their global spread. Various robotic platforms have been developed and tested in the preclinical and clinical settings to demonstrate their efficacy and safety. In this article, we review the publications on technology and elucidate their advantages and existing challenges.

## 1. Introduction

The evolution of endoscopic tissue resection techniques has led to the development of minimally invasive treatments for the most superficial lesions of the gastrointestinal tract. This evolution began with the advent of polypectomy in the late 1960s, followed by endoscopic mucosal resection (EMR) in the 1980s, and endoscopic submucosal dissection (ESD) in the late 1990s. As we transitioned into the 21st century, the concept of natural orifice transluminal endoscopic surgery (NOTES) was introduced. NOTES is a surgical procedure characterized by the introduction of a flexible endoscope through natural body openings, such as the oral cavity, rectum, or vagina, and the creation of an artificial perforation on the wall of an internal hollow organ, enabling minimally invasive surgical interventions within the extraluminal cavities [1,2,3,4,5,6,7,8,9]. This approach can potentially reduce the pain and recovery time associated with traditional surgical methods. The research and development of NOTES has expanded globally. Concurrently, research on the endoscopic full-thickness resection (EFTR) technique, a long-standing aspiration technique for endoscopists, has begun to explore its clinical feasibility for the removal of tumors involving deeper layers as an eventual application in endoscopic tissue resection. EFTR involves the endoscopic resection of the full-thickness gut wall and secure closure of the resultant defect [10,11,12,13,14,15,16,17,18,19,20]. In performing these complex endoscopic surgical techniques, there are procedural limitations in arbitrarily manipulating tissues using off-the-shelf catheter-based apparatuses via the accessory channel of a flexible endoscope. To comprehensively improve the procedure efficiency and ensure the safety of the highly complex endoscopic interventions, the development of “dual-arm” devices with articulation functions, in other words a robotic platform, has been explored; the aim is to enable the triangulation of surgical tools within a remote operative field to be achieved at the level equivalent to laparoscopic surgery. Various endoscopic robotic platforms have been tested in preclinical and clinical settings (Table 1). In this article, we review promising robotic technologies and describe their characteristics as well as the results of pilot clinical trials.

## 2. Technological Features

### 2.1. The Direct Drive Endoscopic System (DDES)

The DDES (Boston Scientific Corp., Natick, MA, USA) is a manual triangulation platform equipped with two arms at the distal end of an overtube (Figure 1A) [21]. The over-tube is 22 mm in diameter and composes three lumens which accommodate a small-caliber endoscope (less than 6 mm in diameter) and two optional tools (less than 4 mm in diameter). Two physicians were required to perform the operation: one for the manual manipulation of the articulated arms and the other for the manipulation of the endoscope. This system has seven degrees of freedom (DOFs). This technology has been tested in both ex vivo and in vivo animal models. This system allows endoscopists to securely close full-thickness defects in the gut wall using a regular curved needle and surgical sutures [21].

### 2.2. EndoSAMURAI

EndoSAMURAI (Olympus Medical Systems, Tokyo, Japan) is a manual triangulation endoscopy system with two 2.8 mm articulated arms and five DOFs mounted on the tip of a 15 mm caliber endoscope with an accessory channel (Figure 1B) [22,24]. Two physicians are required to manipulate the system: one for maneuvering the endoscope and another for manipulating the two articulated arms with independent dual hand-levers on a remote console. A feasibility study of EFTR using this platform was conducted using both ex vivo and in vivo porcine models [22]. It enables more effective tissue manipulation and hastened most of the procedural processes of EFTR compared to EFTR using a double-channel endoscope. Transgastric small intestine resection included manipulation, the devascularization of the mesentery, and stitching the gastric entry point closed using a curved needle and sutures in three porcine models without obvious intraoperative adverse events [25]. The anastomotic leak pressure was measured at necropsy and compared with anastomoses created with surgical staplers and hand suturing. The median time to complete the anastomosis was 41 min (range, 31–65 min). The median leak pressures recorded for the three methods were 14 mmHg (range, 8–33 mmHg), 25 mmHg (range, 13–59 mmHg), and 15 mmHg (range, 12–31 mmHg). In the in vivo study, the median total procedure time from resection to anastomosis was 110 min (range, 90–125 mmHg), and the median leak pressure was 53 mmHg (range, 19–68 mmHg). This report highlights the specific advantages of this platform for surgeons with expertise in laparoscopic techniques but a lack of sufficient experience in complex flexible endoscopic interventions, attributed to the design of the console resembling laparoscopic operation settings.

### 2.3. Master and Slave Transluminal Endoscopic Robot (MASTER)

MASTER (EndoMaster Pte Ltd., Singapore) was developed for the endoscopic submucosal dissection of early-stage gastric cancer by a group at the University of Singapore and Nanyang Technological University [26,27,28] (Figure 1C). MASTER is a master-slave robotic endoscopy system that uses an ergonomic interface to assimilate laparoscopic surgery settings and enables bimanual tasks to be performed intuitively. The system has two arms: mounting grasping forceps and an electrocoagulation hook, individually delivered through the dual instrumental channels of a wide-caliber endoscope. Two operators are required to manipulate the system: an endoscopist, who maneuvers the endoscope to access the operative field, and a surgeon, who operates two robotic arms on a remote master console using a dual-hand lever with haptic feedback.

The technological feasibility of five ex vivo porcine stomach models was reported in 2010 [26]. Following an in vivo porcine study [27], a pivotal pilot clinical trial of the first robotic ESD in humans was conducted in five cases of early-stage gastric cancer in 2012. [28]. R0 resection was achieved in all patients. The mean procedure time was 39 min (range: 26–68 min) for tumors with a median size of 20 mm. No adverse events were observed by postoperative day 30. Additionally, a series of animal studies have demonstrated the feasibility of EFTR, esophageal ESD, and colon ESD [29,30,31]. Transgastric hepatic wedge resection was performed in two porcine models. The total procedure time and resection time were 9.4 min (range, 8.5–10.2 min) and 7.1 min (range, 6–8.2 min), respectively [32] The first generation of the system could not exchange devices. The system was updated (EndoMaster EASE), had three working channels, and two robotic forceps were retrievable [23]. A clinical trial on colorectal ESD using the EndoMaster EASE began in 2020 and is ongoing (NCT04196062).

### 2.4. Endoluminal Assistant for Surgical Endoscopy (EASE)

EASE (ICube Laboratory, Strasbourg, France) was developed as a second-generation Single-access Transluminal Robotic Assistant for Surgeons (STRAS) [33,34]. The system was originally developed as a manually driven triangulation platform equipped with two arms at the tip of an endoscope, known as Anubiscope (Karl Storz Endoskope, Tuttlingen, Germany), and was robotized in later models [35,36]. The tip of the endoscope has a splitting pointed shell composing two articulated device delivery tubes. By opening the shell, the instrumental tubes were split into two different directions, enabling device delivery in the triangulation position. The system was tested in pig models and cadavers, and eventually used for transvaginal cholecystectomy in humans [37,38]. Although the Anubiscope requires two physicians to operate, the scope and devices of the EASE system can be manipulated by a single operator. The system features 10 DOFs, including two robotic instrument channels with a diameter of 4.3 mm each, and an accessory channel for conventional endoscopic instruments with a diameter of 3.2 mm. The physician can remotely control the movement of the endoscope by using levers and four-way thumb switches on the master console. The EASE/STRAS system was developed specifically for colorectal procedures [33,39]. Out of 18 cases of colonic ESD performed in animal models using this robot, complete resection was successfully achieved in 12 attempts. The mean resected specimen size was 18.2 ± 9.8 cm^2^ and the mean total procedure time was 73 ± 35.5 min. Five ESDs resulted in perforation, and system failure occurred in four attempts [39]. The EAES system accomplished a large colorectal ESD significantly faster than ESD using an Anubiscope [33].

### 2.5. Endoscopic Therapeutic Robot System (ETRS)

The Endoscopic Therapeutic Robot System (ETRS) is a master-slave type robotic endoscopy system that allows articulated forceps, an articulated needle knife, fluid injection with a needle, and a regular flexible endoscope to be remotely operated on a multitasking console [40]. The insertion, withdrawal, up-down angulation, rotation, suction, and air insufflation of the endoscope can also be performed without holding the endoscope itself [41,42,43]. Although the system is still at the conceptual modeling level, it has been demonstrated that gastric ESD can be remotely completed without any on-site assistance in ex vivo pig stomach models [40].

### 2.6. i^2^ Snake Robotic Platform

The i^2^ snake robotic platform (Hamlyn Centre for Robotic Surgery, London, UK) was initially designed for head and neck surgery in otolaryngology (Figure 2). This robotic platform is an enhanced version of the iSnake robotic system, specifically developed for single-port surgery [44]. The main body of this endoscopic robot consists of 13 stainless-steel components with 12 rolling joints, providing 7 DOFs. Four instrumental channels with a diameter of 4 mm are available for a camera with illumination, surgical tools, and suction/irrigation. The master interface, which is controlled by a single operator, consists of a handheld gripper and a set of three pedals. Although this platform is also expected to be applicable to endoluminal procedures such as ESD or peroral endoscopic myotomy, in vivo studies or clinical data have not yet been reported [44].

### 2.7. Flex Robotic System

The Flex Robotic System (Medrobotics Corporation, Raynham, MA, USA) was initially designed for head and neck surgery (Figure 3). The system was upgraded to provide adequate insufflation for colorectal procedures, enabling access to areas 25 cm distal to the anal verge [45,46]. This system was approved for colorectal procedures by the U.S. Food and Drug Administration (FDA) in 2017 as the first endoscopic robotic platform approved for clinical use. The robot consists of a flexible insertion tube, camera, and two working channels. The instruments for grasping, cutting, and suturing can be delivered through the channel and can be operated by a single surgeon. The monitor also has three-dimensional vision. The effectiveness of this system for colonic ESD was evaluated and compared with that of conventional ESD in ex vivo bovine models [46]. A total of five novice endoscopists performed two cases of robotic ESD and conventional ESD. En bloc resection was achieved in all cases in the robotic ESD, but only in 50% of cases in the conventional ESD (*p* < 0.0001). Robotic ESD also demonstrated a significantly shorter procedure time (34.1 vs. 88.6 min, *p* = 0.001) and a lower perforation rate (30% vs. 60%, *p* = 0.002). These results indicate that the learning curve of the system is shorter than that of conventional ESD [46]. In addition to ESD, the technical feasibility of Zenker’s diverticulum treatment and EFTR experiments has been explored in in vivo animal models [45]. The feasibility of four types of NOTES procedures—transanal mesorectal excision (taTME), transvaginal hysterectomy, transvaginal salpingo-oophorectomy, and transcecal appendectomy—was sequentially evaluated in a cadaver [47]. The taTME and transcecal appendectomy took 57 min and 24 min, respectively. The ovary and fallopian tube were transvaginally removed in 13.5 min. Although laparoscopic assistance was required, transvaginal hysterectomy was performed in 78 min.

### 2.8. Revolute Joint-Based Auxiliary Transluminal Endoscopic Robot (REXTER)

REXTER is a detachable robotic platform developed for ESD designed to facilitate mucosal tenting and submucosal tissue dissection (Figure 4A) [48]. The system comprises a robotic arm, an actuator enclosure, and a control interface. The robotic arm with four DOFs is mounted on the tip of the endoscope. An assistant physician operates the interfacial console. In a comparative study between robotic and conventional ESD ex vivo, robotic ESD achieved a lower perforation rate among novice operators (1/10 vs. 6/10) [48].

### 2.9. Portable Endoscopic Tool Handler (PETH)

The PETH (KAIST; Korea Advanced Institute of Science and Technology, Daejeon, Korea) is a robotic endoscopic platform composed of a robotic arm, master device, graphics simulator, and motor pack (Figure 4B). The robotic arm has two DOFs attached to a regular endoscope. There are two variations: single- and dual-arm systems. Both the number and direction of robotic arms can be changed as required. The arm has a hollow structure with a 2.8 mm channel accommodating most of the currently available flexible devices. The motion of each robotic arm is controlled using the thumb stick of the master console. Twenty ESD procedures were performed using the PETH and compared with 15 conventional ESD procedures. Complete resection was performed in all patients, and no perforations were observed. PETH-ESD achieved a significantly lower blind dissection rate (0 vs. 20%, *p* < 0.001) and faster dissection speed (122.3 ± 76.5 vs. 47.5 ± 26.9 mm^2^/min, *p* < 0.001) [49].

### 2.10. K-FLEX

The K-FLEX (KAIST, Daejeon, Korea) consists of surgical instruments, an overtube, two robot arms, and a master console fitting with an outer diameter of 17 mm. The system has 14 DOFs, with two instruments and an overtube [50]. The system has an appropriate size for insertion through natural orifices, and the robot arm has sufficient force for the traction and incision of the mucosa, as confirmed by payload measurements. The technical feasibility of ESD was demonstrated by ex vivo experiments. The visibility of the operative field hindered by robotic arms is a challenge that necessitates further miniaturization and design modifications.

### 2.11. Endoluminal Surgical (ELS) System

An Endoluminal Surgical System (ELS; Colubris MX, Inc., Houston, TX, USA) was developed for transanal surgery (Figure 5) [49]. The ELS is designed as an open master console with high-definition 2D optics. The Colubriscope is the main operating head, with a 22 mm diameter composed of a videoscope channel, two insufflation channels, two surgical tool channels, and a biopsy and suction channel. Although the motion of the insertion and withdrawal of the overtube needs to be operated by an on-site assistant using a touch screen, other functions can be controlled by a surgeon at the console using dual hand pieces and foot pedals. Robotic instruments, such as a 6 mm robotic grasper, hook cautery tip, scissor tip, and needle drivers, can be utilized through instrument channels. The tips of these devices were equipped with multi-articulated joints, allowing for seven DOFs. The technical feasibility of ESD was tested by a total of 20 colonic lesions in an ex vivo porcine model by a surgeon [49]. All lesions, 25–35 mm in size, were successfully resected en bloc without adverse events. The mean procedure time (±standard deviation; SD) was 18.41 ± 14.15 min. Nine lesions were sutured with 4-0 verbed nylon sutures. The mean suturing time (±SD) was 27.89 ± 10.07 min. In this study, the latter 10 lesions had significantly shorter procedure times than the former 10 lesions (*p* = 0.007). The findings indicate that the learning curve for this system can reach a sufficient level in a short period and in a few cases. Clinical trials on ESD and bariatric surgery are underway.

### 2.12. Flexible Auxiliary Single-Arm Transluminal Endoscopic Robot System (FASTER)

The FASTER robot is attached to the tip of a flexible endoscope (Figure 4C) [51]. The robotic system is composed of a robotic arm, drive housing, and manipulating console. This robot was primarily designed to provide mucosal traction during ESD. The tissue-grasping arm has three DOFs: down bending, right/left bending, and open/close. Two physicians are required for the operation: one physician for manipulating the endoscope and another physician for operating the robotic arm. In vivo animal testing for gastric ESD [51] demonstrated that robotic ESD significantly hastens tissue dissection. However, no significant difference was observed in areas subjected to gravitational traction, such as lesions on the anterior wall. In a randomized controlled trial with six pigs (four for the esophagus and four for the stomach in each pig, 48 lesions in total) [52], robotic ESD resulted in a significantly shorter procedural time (18.8 vs. 32.8 min; *p* < 0.001) and notably fewer incidents of muscularis propria injury (*n* = 6 vs. 21; *p* = 0.018) than conventional ESD. The benefits of this system are more evident for esophageal ESD than for gastric ESD.

## 3. Discussion

Endoscopic procedures such as ESD and EFTR have undergone significant advancements, largely attributable to the evolution of therapeutic devices and the enhancement of endoscopists’ technical proficiency. The flexible endoscope, initially designed for diagnostic purposes including biopsy, was fundamentally intended for single-device usage. Complex endoscopic therapeutic procedures require advanced technical skills. To overcome these challenges and augment the efficacy and safety of the procedure, the development of endoscopic devices, including traction methods and suturing devices, has progressively advanced [53,54,55,56,57]. Simultaneously, notable progress has been made in the development of flexible endoscopic robotic platforms, particularly for NOTES, which facilitates bimanual manipulation. The primary objective of developing flexible endoscopic robots is to address the issues related to triangulation and countertraction, thereby enabling the execution of procedures involving complex maneuvers. These endoscopic robots offer an enhanced visualization of the surgical field and improved precision in surgical maneuvers, thereby possessing the potential to manage complex procedures and reduce the learning curve [37,46]. In this study, we reviewed the characteristics of each robotic platform and its application in clinical or animal trials. As the required functionalities of the robots would vary with each endoscopic procedure, we discuss them based on the endoscopic procedures in which the integration of endoscopic robots is anticipated.

### 3.1. Applications for ESD, EFTR, and NOTES

#### 3.1.1. ESD

Since its inception in Japan in the 1990s, ESD for gastrointestinal tumors has gained widespread acceptance owing to its high therapeutic effectiveness, including the *en bloc* resection rate [58,59,60]. A stable view of the submucosal layer is essential for the success of ESD. However, stabilizing the field of vision is difficult with flexible endoscopic treatment [61,62,63,64,65]. Traction methods have been developed to address this issue, which have contributed to a safe procedure and decreased procedure time. However, some cases of ESD, such as lesions with fibrosis, poor scope operability, and large lesions, require advanced technical skills [66]. Traction methods have been developed to address this issue, which have contributed to a safe procedure and decreased procedure time. However, some cases of ESD, such as lesions with fibrosis, poor scope operability, and large lesions, require advanced technical skills [28]. All five procedures were performed without any complications. However, this system was only used to dissect the submucosal layer, and the marking and circumferential incisions were made using a conventional endoscope. To address this problem, a second-generation MASTER system (EndoMASTER EASE) was developed [23]. A clinical trial of colorectal ESD using this platform has been initiated, and its outcomes are anticipated. The Flex robotic and ELS systems are also expected to be practically utilized in ESD. These platforms possess advanced capabilities for deep recognition during procedures and allow for precise manipulation without interference between devices. The difference between these platforms is that in the Flex system, manual manipulation is necessary for devices inserted through the working channels, whereas the ELS system enables the robotic control of these devices. In addition, the seven DOFs of the ELS system provide the flexible motion of the devices, including grasping, knife, and scissor-type forceps. The results of clinical trials using these systems are expected.

#### 3.1.2. EFTR

The use of laparoscopic and endoscopic cooperative surgery (LECS) and EFTR for small gastrointestinal stromal tumors (GISTs) has increased in recent years [67]. LECS involves a circumferential incision of the tumor with a flexible endoscope, allowing for minimal resection margins. The surgical closure of the defect facilitates reliable and safe tumor removal. However, this method requires the use of laparoscopic ports, necessitating skin incisions and mesenteric processing. In contrast, EFTR is a minimally invasive treatment that involves tumor resection and defect closure using only a flexible endoscope. However, EFTR faces several challenges, including difficulties in maintaining the field of view due to air leakage, especially for anterior wall lesions, and frequent struggles with defect closure, leading to leakage [19,68]. Although full-thickness closure devices such as Overstitch (Apollo Endosurgery, Austin, TX, USA) [53,54], the double-armed bar suturing system (Zeon Medical, Tokyo, Japan) [55], and the over-the-scope clip (Ovesco Endoscopy, Tübingen, Germany) [56,57] have been reported, they also require technical expertise for effective use. Ensuring reliable closure is also a crucial aspect of the robot used in the EFTR. The increase in the DOFs of a robot’s wrist allows for finer and more precise movements, enabling the emulation of natural hand movements with greater accuracy. This provides significant benefits when performing complex procedures such as defect suturing. Several robotic platforms, such as EndoSAMURAI, MASTER, and Flex robotic systems, were examined in experiments designed for the EFTR [22,30,44]. The Flex Robotic system successfully performed all EFTR procedures, from ESD to defect suturing after resection, in an in vivo animal trial. The system offers various suture patterns with a needle and thread using bimanual control and flexible arms. Triangulation also helps in performing such delicate procedures. Further validation of robotic endoscopy is necessary for the practical application of EFTR.

#### 3.1.3. NOTES

NOTES has seen worldwide expansion in research following a report on transgastric peritoneoscopy in porcine models in 2004 [1]. Various approaches from different routes, such as transgastric cholecystectomy, splenectomy [3,6,8], and transvaginal distal pancreatectomy [9], have been demonstrated. Following these studies, there was a surge in NOTES research; however, there were many challenges, including the lack of reliable closure methods for gastrointestinal defects and the complexity of the surgical techniques. The development of flexible endoscopic robots is essential for the future development of NOTES. MASTER was initially developed for NOTES. Transgastric segmental hepatectomy in porcine models using this system was performed without complications and in a relatively short procedure time, although the closure of the gastric defect was not performed [32]. The Flex robotic system for the colorectum successfully demonstrated transvaginal NOTES experiments, such as transvaginal hysterectomy and transvaginal unilateral salpingo-oophorectomy [47]. This system has shown the potential for flexible endoscopes to be effective in NOTES, even in anatomically circuitous paths. Although many robots are being researched and developed for NOTES, few studies have reported their efficacy. The advantage of flexible endoscopes is their ability to navigate through curvatures such as from the mouth to the pharynx or from the anus to the intestines. However, balancing flexibility with operability when creating such robots is technically difficult. Therefore, the range that can be reached by the scope is limited. To minimize the incision size of the gastrointestinal orifice, it is necessary to reduce the outer diameter of the endoscope as much as possible, while maintaining the gripping force of the arms and DOF, which makes its development more challenging [50]. Enhanced manipulability, involving an increase in the DOF and a greater number of components and instruments, may complicate the reduction in model size. There are still many challenges to the realization of NOTES in clinical practice.

### 3.2. Considerations and Perspectives

Regarding early cancer detection and less invasive treatment, the role of flexible gastrointestinal endoscopy has been extremely significant. Flexible robotic platforms are expected to simplify and standardize surgical techniques, which have traditionally relied heavily on the skills of endoscopists, and reduce the learning curve. Currently, robotic platforms like EndoMaster and the ELS system are undertaking clinical trials mainly for the ESD technique, which is more advanced and widespread than EFTR and NOTES. The successful clinical application of the ESD technique with a robotic platform could pave the way for its use in more complex procedures such as EFTR and NOTES. In particular, the NOTES procedure faces many challenges with existing devices, making the development of new instruments like robotic endoscopes crucial.

In contemporary medicine, the development and integration of Artificial Intelligence (AI) have become indispensable. Recent advancements in AI in gastrointestinal endoscopy have primarily focused on ensuring the uniform quality of examinations. Most AI systems are designed to provide lesion detection functions (Computer-Aided Detection, CADe) and lesion diagnosis support (Computer-Aided Diagnosis, CADx) based on deep learning, particularly in the field of colonoscopy [69]. AI holds the potential to assist in more intricate tasks, such as providing procedural guidance, executing semi-automated device maneuvers, and aiding in surgical decision-making through automated analysis and the interpretation of computer images, as autonomous robots are being tested in surgical fields. It is expected that flexible robotic endoscopy platforms will also be equipped with AI capable of controlling endoscopic motion and providing anatomical information on the screen, leading to safer and higher-quality minimally invasive treatments.

Despite the complexities involved in developing therapeutic robotic endoscopy platforms that can move through narrow lumens toward the depths of the gastrointestinal tract and perform delicate tasks, they could trigger technological innovations similar to those brought about in surgical operations by robotic systems like the da Vinci, benefiting many people.

## 4. Conclusions

Most robotic endoscopic systems remain in various phases of experimental validation and ongoing development, or are progressing towards market availability. Introducing flexible robotic endoscopy platforms remains challenging because of their size, structural complexity, and cost-effectiveness. It is expected that these issues will be resolved, and flexible endoscopic robots will be clinically introduced, leading to increased procedural safety and decreased procedural difficulty, benefiting a large number of cases.

## Figures and Tables

**Figure 1 diagnostics-14-00595-f001:**
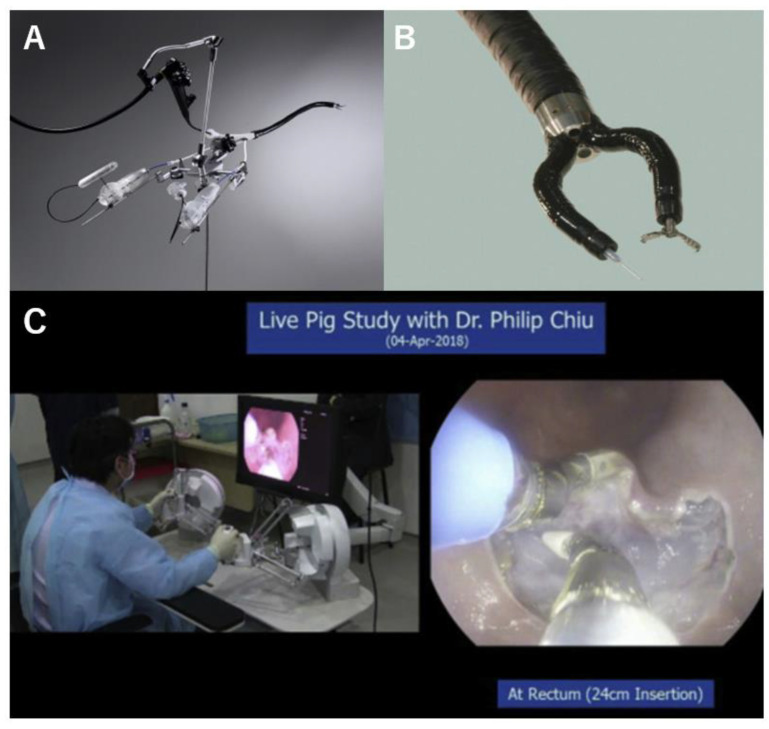
(**A**) Direct-drive endoscopic system [21]. (**B**) EndoSAMURI [22] (**C**) Colonic endoscopic submucosal dissection using EndoMaster EASE. EASE, Endoluminal Assistant for Surgical Endoscopy [23].

**Figure 2 diagnostics-14-00595-f002:**
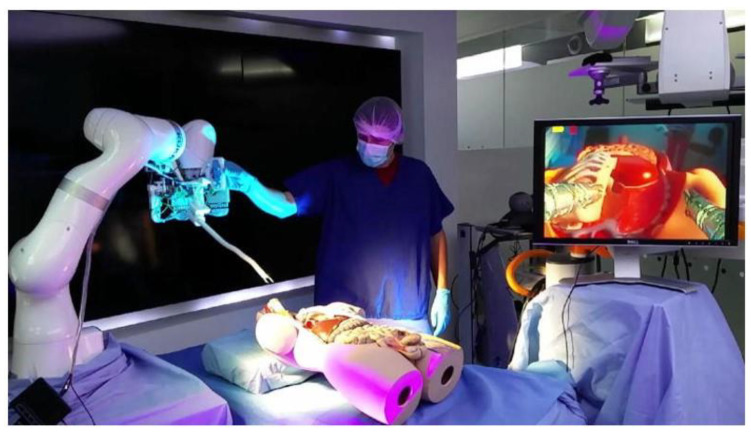
i^2^ Snake Robotic Platform for ENT surgery [44].

**Figure 3 diagnostics-14-00595-f003:**
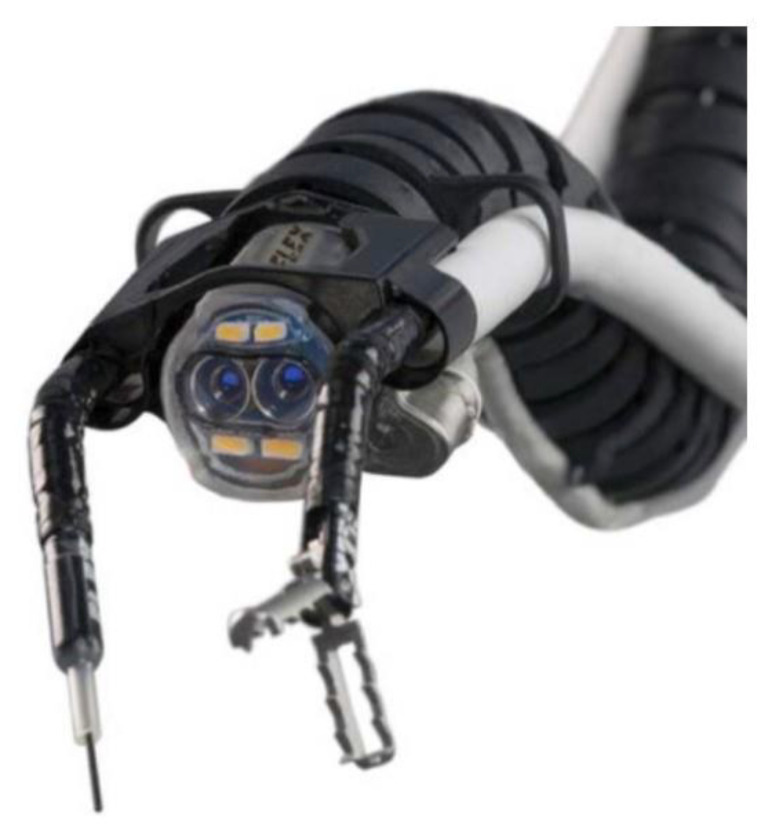
Flex robotic system [46].

**Figure 4 diagnostics-14-00595-f004:**
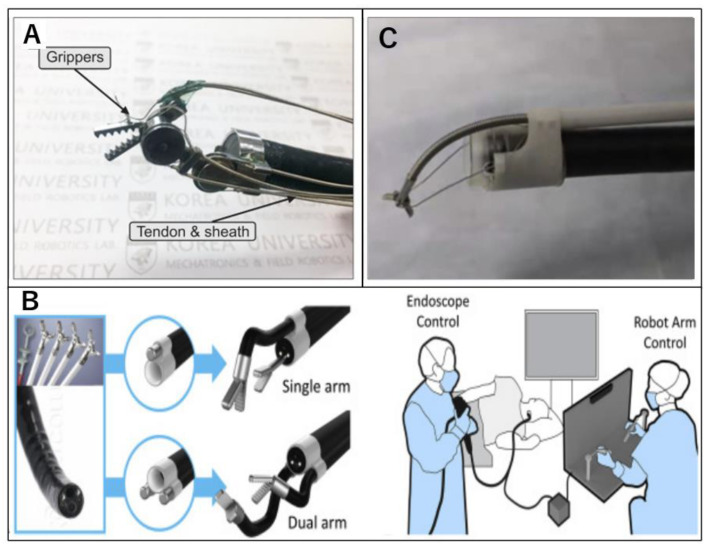
(**A**) Revolute joint-based auxiliary transluminal endoscopic robot. (**B**) Portable endoscopic tool handler. (**C**) Flexible auxiliary single-arm transluminal endoscopic robotic system.

**Figure 5 diagnostics-14-00595-f005:**
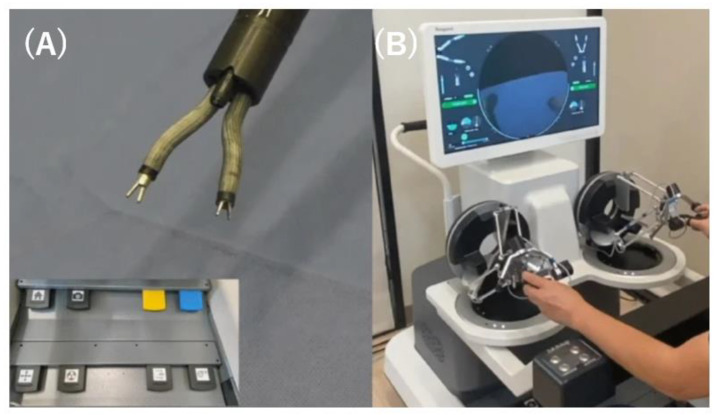
Endoluminal surgical system. (**A**) Robotic arms and foot pedals that control the camera and scope positions [49]. (**B**) the graspers used for operating the robotic instruments

**Table 1 diagnostics-14-00595-t001:** Characteristics and status of flexible endoscopic robots listed in ascending order of publication year for gastrointestinal use.

Device	Operator	Degree of Freedom	Outer Diameter (mm)	Clinical Trial		Approval
The direct drive endoscopic system (DDES)	Double	7	22	EMR NOTES	Animal in vivo	None
EndoSAMURAI	Double	5	15	EFTR NOTES	Animal in vivo	None
Master and slave transluminal endoscopic robot (EndoMaster EASE)	Double	10	N/A	ESD EFTR NOTES	Animal in vivo Human	None
Endoluminal Assistant for Surgical Endoscopy (EASE)	Double	10	18	ESD	Animal in vivo	None
Endoscopic therapeutic robot system (ETRS)	Single	N/A	N/A	ESD	Animal ex vivo	None
i^2^ snake robotic platform	Single	7	16	None	None	None
Flex robotic system	Single	N/A	18	ESD EFTR NOTES	Animal in vivo Human	FDA and CE
Revolute joint-based auxiliary transluminal endoscopic robot (REXTER)	Double	4	N/A	ESD	Animal ex vivo	None
Portable endoscopic tool handler (PETH)	Double	2	N/A	ESD	Animal ex vivo	None
K-FLEX	Single	14	17	ESD	Animal ex vivo	None
Endoluminal surgical system (ELS)	Single	7	22	ESD	Animal ex vivo	None
Flexible Auxiliary Single-arm Transluminal Endoscopic Robot system (FASTER)	Double	3	N/A	ESD	Animal in vivo	None

EMR, endoscopic mucosal resection; ESD, endoscopic submucosal dissection; EFTR, endoscopic full-thickness resection; FDA, Food and Drug Administration; CE, European Conformity; N/A, not applicable.

## Data Availability

Not applicable.

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
