# Peer review of "Robotic Platforms for Therapeutic Flexible Endoscopy: A Literature Review"

_diagnostics, 2024, doi:10.3390/diagnostics14060595_

Round 1

Reviewer 1 Report

Comments and Suggestions for Authors

I would like to congratulate you on the interesting review. I would like to Discussions a table with comparisons of the results after ESD, EFTR and NOTES. At conclusions, I would like you to focus on which technique you think has the greatest future to be implemented in clinical practice after the preliminary results

Author Response

Response)

Thank you for your important comment.

As you mentioned, we have added the following sentences of our opinion for the future perspectives in the new section “Considerations and Perspectives ".

Currently, robotic platforms like EndoMaster and ELS system are undertaking clinical trials mainly for the ESD technique, which is more advanced and widespread than EFTR and NOTES. Successful clinical application of the ESD technique with a robotic platform could pave the way for its use in more complex procedures such as EFTR and NOTES. In particular, the NOTES procedure faces many challenges with existing devices, making the development of new instruments like robotic endoscopes crucial.

Reviewer 2 Report

Comments and Suggestions for Authors

The manuscript is very well-written. I have very few comments: 

To further improve the manuscript, could the authors please include a section on how artificial intelligence can increase the autonomy of endoscopy by contributing to automated processing and interpretation of computer images? 

Also, in the Conclusion section, it would be helpful to may be include a "Considerations and Perspectives" section?

Looks good otherwise. 

Author Response

Response)

Thank you for your important comments. As you suggested, we have added the new section “Considerations and Perspectives” in the discussion section and included sentences referring to artificial intelligence in it, as follows.

3.2. Considerations and Perspectives

Regarding early cancer detection and less invasive treatment, the role of flexible gastrointestinal endoscopy has been extremely significant. Flexible robotic platforms are expected to simplify and standardize surgical techniques, which have traditionally relied heavily on the skills of endoscopists, and reduce the learning curve. Currently, robotic platforms like EndoMaster and ELS system are undertaking clinical trials mainly for the ESD technique, which is more advanced and widespread than EFTR and NOTES. Successful clinical application of the ESD technique with a robotic platform could pave the way for its use in more complex procedures such as EFTR and NOTES. In particular, the NOTES procedure faces many challenges with existing devices, making the development of new instruments like robotic endoscopes crucial.

In contemporary medicine, the development and integration of Artificial Intelligence (AI) have become indispensable. Recent advancements in AI in gastrointestinal endoscopy have primarily focused on ensuring uniform quality of examinations. Most AI systems are designed to provide lesion detection functions (Computer-Aided Detection, CADe) and lesion diagnosis support (Computer-Aided Diagnosis, CADx) based on deep learning, particularly in the field of colonoscopy. [71] AI holds the potential to assist in more intricate tasks, such as providing procedural guidance, executing semi-automated device maneuvers, and aiding in surgical decision-making through automated analysis and interpretation of computer images, as autonomous robots are being tested in surgical fields. It is expected that flexible robotic endoscopy platforms will also be equipped with AI capable of controlling endoscopic motion and providing anatomical information on the screen, leading to safer and higher-quality minimally invasive treatments.

Although the complexities involved in developing therapeutic robotic endoscopy platforms that can move through narrow lumens toward the depths of the gastrointestinal tract and perform delicate tasks, they could trigger technological innovations similar to those brought about in surgical operations by robotic systems like the da Vinci, benefiting many people.

  1. Sumiyama K, Futakuchi T, Kamba S, Matsui H, Tamai N. Artificial intelligence in endoscopy: Present and future perspectives. Dig Endosc. 2021 Jan;33(2):218-230. doi: 10.1111/den.13837. Epub 2020 Nov 3. PMID: 32935376.